# Ovariectomy Exacerbates Acute Ethanol-Induced Tachycardia: Role of Nitric Oxide and NMDA Receptors in the Rostral Ventrolateral Medulla

**DOI:** 10.3390/ijms24065087

**Published:** 2023-03-07

**Authors:** Jiro Hasegawa Situmorang, Hsun-Hsun Lin, Md Sharyful Islam, Chih-Chia Lai

**Affiliations:** 1Master and PhD Programs in Pharmacology and Toxicology, School of Medicine, Tzu Chi University, Hualien 970374, Taiwan; 2Center for Biomedical Research, National Research and Innovation Agency (BRIN), Cibinong 16915, Indonesia; 3Department of Physiology, School of Medicine, Tzu Chi University, Hualien 970374, Taiwan; 4Department of Pharmacology, School of Medicine, Tzu Chi University, Hualien 970374, Taiwan

**Keywords:** estrogen, ethanol, heart rate, nitric oxide, NMDA, RVLM, nucleus ambiguus

## Abstract

Ethanol consumption influences cardiovascular functions. In humans, acute consumption of ethanol causes dose-dependent tachycardia. Our previous study showed that ethanol-induced tachycardia might involve decreased nitric oxide (NO) signaling in the brain’s medulla. NMDA receptors, another important target of ethanol, are one of the upstream signals of nitric oxide. Reports showed the modulation of NMDA receptor function by estrogen or estrogen receptors. The present study aims to examine the hypothesis that depletion of estrogen by ovariectomy (OVX) might modulate ethanol-induced tachycardia by regulating NMDA receptor function and NO signaling in the cardiovascular regulatory nucleus of the brain. Ethanol (3.2 g/kg, 40% *v*/*v*, 10 mL/kg) or saline (10 mL/kg) was administered by oral gavage in sham or OVX female Sprague-Dawley (SD) rats. The blood pressure (BP) and heart rate (HR) were measured using the tail-cuff method. The levels of phosphoserine 896 of the GluN1 subunit (pGluN1-serine 896) and NMDA GluN1 subunits (GluN1) were determined by immunohistochemistry. The expressions of nitric oxide synthase (NOS) and estrogen receptors in the tissue were measured by Western blotting. Nitric oxide contents were measured as total nitrate-nitrite by colorimetric assay kit. In a 2-h observation, there was no significant change in BP between the saline and ethanol groups. However, compared with saline, ethanol caused an increase in HR (tachycardia) in sham control or OVX rats. Interestingly, ethanol produced more significant tachycardia in the OVX group than in the sham control group. Nitric oxide levels were lower in the area of the rostral ventrolateral medulla (RVLM) 60 min following ethanol administration in OVX compared with sham control, without significant changes in the expression of NOS and estrogen receptors (ERα and ERβ). In addition, a decrease in the immunoreactivity of pGluN1-serine 896, without significant changes in GluN1, was found in neurons of RVLM 40 min following ethanol administration in OVX compared with sham control. Our results suggest that depletion of estradiol (E2) by OVX might exacerbate the tachycardia following ethanol administration, the underlying mechanism of which might be associated with decreased NMDA receptor function and NO level in the RVLM.

## 1. Introduction

Ethanol is one of the most commonly abused substances worldwide, and it can cause destructive health consequences when over-intake ethanol, as in binge drinking. Once consumed, it will be absorbed throughout the body, including the brain. Excessive ethanol consumption comes with disease progression and risk of cardiovascular disease, which affects the normal function of blood pressure (BP) and heart rate (HR) [1,2]. A prospective study in humans revealed that acute ethanol consumption is highly associated with sinus tachycardia, where higher breath ethanol concentration resulted in a higher prevalence of sinus tachycardia [3].

Estrogen is reportedly involved in regulating the cardiovascular system. Activation of estrogen receptors (ER) has cardioprotective effects [4]. Short-term estrogen treatment was also shown to help lower BP in preeclampsia by modulating plasma oxidative stress [5]. In ovariectomized rats that resemble menopause in women, estrogen treatment enhanced baroreflex control and restored the attenuated cardiac vagal afferent reflex of HR [6,7].

Several studies reported that the ethanol acting in the central nervous system (CNS) is responsible for its effects on cardiovascular function. Previous studies found that N-methyl-D-aspartate (NMDA) inhibition in the rostral ventrolateral medulla (RVLM) and gamma-Aminobutyric acid (GABA) potentiation in nucleus tractus solitarii (NTS) might mediate the inhibitory effects of ethanol on baroreflex bradycardia [8,9]. Our previous studies showed that acute ethanol-induced hypotension in anesthetized rats might involve GABA/(NMDA/nitric oxide (NO) in the RVLM [10,11]. Furthermore, the reduction of nNOS expression and NO levels at the rostroventral medulla might play a role in the tachycardia induced by subacute administration of ethanol using telemetry method in conscious female rats [12].

In conscious female rats, ovariectomy (OVX) abolished the hypotensive effects of acute administration of ethanol; treatment of OVX rats with estradiol (E2) restored the hypotensive and sympathoinhibitory responses to ethanol [13]. A study showed that elevated NO level in ethanol-induced hypotension in the NTS is estrogen-dependent [14]. Ethanol-evoked myocardial oxidative stress and dysfunction are also estrogen-dependent [15]. Despite these pieces of knowledge, the mechanistic role of estrogen in regulating ethanol-induced tachycardia remains obscure. NMDA receptors, important targets of ethanol, are the upstream signals of nitric oxide [16]. Protein phosphorylation is an important mechanism for regulating NMDA receptor function [17]. Several studies reported estrogen’s role in regulating NMDA receptor function and the phosphorylation of NMDA receptor subunits in neurons [18,19,20]. The present study aims to examine the hypothesis that depletion of estrogen by OVX might modulate tachycardia following acute ethanol exposure by regulating NMDA receptors and NO signals in the cardiovascular regulatory nucleus of the brain. We examine changes in the NMDA receptor phosphorylation and NO level in the RVLM and nucleus ambiguus (NA), which controls cardiovascular sympathetic and parasympathetic activity, respectively. Eventually, we showed that the increase in HR elicited by ethanol is higher in OVX rats. There was no significant difference in the expression of nitric oxide synthase (NOS) and estrogen receptors in the RVLM and NA between the sham control and OVX group. OVX rats showed a decrease in nitric oxide levels, predominantly in the RVLM area. E2 supplement attenuated the increased HR and restored the NO levels in RVLM. In addition, a decrease in the immunoreactivity of pGluN1-serine 896, without significant changes in GluN1, GluN2B subunit (GluN2B), and the phosphoserine 1480 of the GluN2B subunit (pGluN2B-serine 1480), was found in neurons of RVLM in OVX rats compared with sham control.

## 2. Results

### 2.1. Ovariectomy Increased Ethanol-Induced Tachycardia, and Estradiol Treatment Attenuated the Tachycardia

We investigated the effect of OVX with or without E2 treatment, a predominant type of estrogen, on mean arterial pressure (MAP) and HR after the acute administration of ethanol. The basal MAP and HR values were measured 30 min before ethanol or saline administration. The basal BP and HR were 108.5 ± 3.3 and 347.9 ± 6.2 (*n* = 6) for sham control rats given saline, 94.3 ± 3.7 and 352.1 ± 8.0 (*n* = 5) for sham control rats given ethanol, 99.5 ± 7.0 and 325.7 ± 6.6 (*n* = 5) for OVX rats given ethanol, 100.9 ± 2.3 and 328.3 ± 5.9 (*n* = 6) for OVX rats daily supplemented with E2 and given ethanol. The basal values were not statistically different among different groups analyzed by one-way ANOVA followed by Tukey’s post hoc test. In a 2-h observation, ethanol administration did not influence MAP in all treatment groups compared to saline administration (Figure 1a,b). However, compared with saline, ethanol caused an increase in HR (tachycardia) in sham control or OVX rats. Interestingly, over time, ethanol produced more significant tachycardia in the OVX group than in the sham control group (Figure 1a). Daily subcutaneous E2 attenuated the tachycardia evoked by ethanol (Figure 1b).

### 2.2. Ovariectomy and E2 Treatment Did Not Alter NOS and ER Expression in RVLM and NA

Neither OVX nor E2 treatment influenced the NOS expression and ERs expression. After two weeks of the OVX or E2 treatment, the expression of neuronal NOS (nNOS), inducible (iNOS), endothelial NOS (eNOS), estrogen receptor alpha (ERα), and estrogen receptor beta (ERβ) remained the same in both RVLM and NA compared to sham control (Figure 2 and Figure 3).

### 2.3. Ovariectomy Decreased NO Levels in the RVLM

Compared to saline, oral gavage of ethanol had no significant effects on the NO levels in the RVLM and NA of the sham rats. However, the NO level was notably lower in the RVLM of the OVX group compared to that in the sham group after ethanol administration. Consequent chronic E2 subcutaneous injection reversed the NO suppression caused by OVX (Figure 4a). In the NA, the OVX group showed a lower but insignificant NO level compared to the sham group. The NO level increased in OVX rats supplied with E2 compared to that in the OVX group (Figure 4b).

### 2.4. Changes in the Immunoreactivity of NMDA Receptor Subunit Phosphorylation in the RVLM

We examined the effects of ethanol on the expression of GluN1, pGluN1-serine 896, GluN2B, and pGluN2B-serine 1480 in the RVLM neurons by immunohistochemistry. Compared to saline, oral gavage of ethanol had little effect on the number of immunoreactive (IR) neurons of the subunits and phosphorylated site on the subunits in the RVLM 40 min following administration. However, the number of neurons in the RVLM area showing immunoreactivity to pGluN1-serine 896 significantly decreased in OVX groups compared to the sham rats following ethanol administration; there was no significant difference in the numbers of the IR neurons of GluN1, GluN2B, GluN2B-serine 1480 between the sham and OVX following ethanol administration (Figure 5).

## 3. Discussion

A human study reported that high ethanol consumption in alcoholic beverages is positively associated with tachycardia [3]. In the present study, we administered rats with ethanol by oral gavage at a dose of 3.2 g/kg (40% *v*/*v*). Our previous study [12] has shown that the blood concentration was around 80–100 mg/dL (0.08–0.1%) 30–60 min following oral gavage of ethanol (3.2 g/kg), which is comparable to levels observed in many human drinkers. Female hormones may play a role in regulating cardiovascular function. In the present study, we tried to elucidate the role of estrogen in mediating HR changes during ethanol exposure in female rats. Our results showed a slight yet insignificant reduction of baseline HR in OVX rats. However, OVX showed more sensitivity to the tachycardia elicited by ethanol. Administering E2 to compensate for the hormonal loss in the OVX group attenuated the tachycardia. Also, we observed that NO levels significantly decreased in the RVLM and, to a lesser extent, in the NA 60 min after ethanol administration when the tachycardia effects were more evident in the OVX rats. Daily treatment with E2 reversed the decreased NO level. In addition, a decrease in the immunoreactivity of pGluN1-serine 896 was found in neurons of RVLM following ethanol administration in OVX rats. The results imply that NMDA receptor function and NO signaling in the RVLM might participate in estrogen regulation of HR during ethanol exposure.

Several studies have reported the involvement of NO in regulating HR. At baseline, neuronal NO synthase (nNOS) knockout mice show higher HR [21]. In RVLM and NA, NO’s signaling in mediating HR changes has been studied. Studies showed that microinjection of NO donors into the RVLM caused sympathoexcitation, resulting in increased BP and decreased HR in conscious animals [22,23]. On the other hand, NO reduction in NA caused a significant increase in HR, whereas NO enhancement caused a significant decrease in HR in anesthetized rats [24]. Our previous study revealed that decreases in nNOS and NO in the medulla might be associated with the worsened tachycardia elicited by subacute ethanol consumption [12]. Studies have proven that OVX could cause changes in NO signaling. For example, in the cerebral blood vessel of OVX rats, the eNOS expression was reduced and accompanied by the reduction of NO level. These effects were reversed by E2 treatment [25]. In cultured brain endothelial cells, another study has shown that E2 could increase NO levels via eNOS, while this effect was abrogated by co-treatment with E2 antagonism [26]. In rat’s alveolar macrophages, OVX has been shown to reduce the spontaneous release of NO [27]. Consistent with these studies, our data show that the NO level in RVLM of OVX rats is lower than that of sham rats following ethanol administration. NO level in NA also shows a lower tendency. We, therefore, suggest lower NO levels found in OVX rats could result from the loss of estrogen. To further clarify this, we try to elucidate the role of E2 in regulating tachycardia during ethanol exposure in OVX rats. By two weeks of E2 treatment right after OVX, the tachycardia effect of ethanol was attenuated compared with OVX rats. As expected, we found the NO level of E2-treated OVX rats to be similar to the NO level of sham rats both in RVLM and NA. Despite the changes in NO levels in OVX and E2 treatment rats, our data showed no difference in the three types of NOS expression. Thus, the regulation by estrogen of ethanol-induced changes in NO levels might result from the alteration in the catalytic activity of NOS rather than the expression of NOS. Given that nNOS is the primary type of NOS in the brain, we speculate that it may be more closely associated with the effect of OVX and ethanol in the current study.

Protein phosphorylation plays a central role in the functional regulation of NMDA receptors [28]. Reports have indicated that estrogen might modulate nociceptive transmission by regulating phosphorylated sites on or the expression of GluN1 and GluN2B subunits in dorsal horn neurons of the spinal cord [29,30,31]. pGluN1-serine 896 and pGluN2B-serine 1480 are substrates for protein kinase C and casein kinase II, respectively. Ethanol alters protein kinase C (PKC) activity and expression in the brain [32]. Activation of PKC pathways by ethanol potentiated NMDA receptor function and the underlying neuronal activity in spinal dorsal horn neurons [33,34]. On the other hand, ethanol might reduce NMDA receptor function, activated by PKC activator, via decreases in PKC activity in primary cultured cerebellar granule cells [35]. PKC phosphorylated pGluN1-serine 896; increases in pGluN1-serine 896 potentiated NMDA receptor activity and the underlying neuronal function [36,37]. The present study showed a decrease in pGluN1-serine 896 following ethanol administration in OVX rats. The depletion of estrogen by OVX likely modulates ethanol effects on pGluN1-serine 896, resulting from an inhibition of PKC signaling or activating a phosphatase. Ethanol inhibition of pGluN1-serine 896 in OVX rats might decreased NMDA receptor function and account, at least partly, for decreased NO levels in the present study. However, changes in many signaling pathways and phosphorylated sites on NMDA receptor subunits might modulate NMDA receptor function. Though our results indicated that pGluN1-serine 896 is an important target in response to OVX potentiation of ethanol-induced tachycardia, we cannot rule out the possibility of the involvement of other phosphorylated sites, such as the phosphotyrosine site, on NMDA receptor subunits in the tachycardia responses. The primary hormones produced by the ovaries include estrogen and progesterone. The specific role of estrogen and/or progesterone in regulating NMDA receptor phosphorylation needs to be further clarified.

We demonstrated in the immunohistochemical study that ethanol administration decreased pGluN1-serine 896 positive neurons in the RVLM of OVX rats compared with sham rats. This result provided evidence that OVX might modulate ethanol effects on NMDA receptor function by regulating NMDA receptor phosphorylation (pGluN1-serine 896). The results would give a possible relationship between NMDA receptor function and NO levels, which might support part of the conclusion. However, it is worth mentioning that the results cannot reveal the causal relationship between NMDA receptor function and NO signals in OVX regulation of ethanol effects. Our findings indicated that PKC-regulated NMDA receptor function (pGluN1-serine 896) and NO levels in the RVLM might contribute to OVX regulation of ethanol-induced tachycardia. Thus, by pharmacological approach, administration of NMDA receptor and NO signals modulators into the RVLM might regulate the ethanol-induced tachycardia, which might further prove the causal relation between NMDA receptor and NO signaling in OVX regulation of ethanol effects.

Literature has shown that the ER subtypes (ERα, ERβ, GPER) are expressed in the brain medulla [38,39]. It has been reported that estrogen regulates NMDA receptor function and NO signaling in the preoptic region of the hypothalamus [40]. Estrogen and its receptors are involved in the neuroprotection in the CNS via the inhibition of NMDA receptor function, the mechanism of which might be associated with the activation of signaling pathways [18,19]. ERβ in the RVLM produced cardiovascular depressive effects probably via activation of NO signaling by non-transcriptional mechanisms [41]. These studies demonstrated that ERs modulate NMDA receptor function and NO signaling. In the present study, there is no significant difference in the expression of two ER types in the RVLM and NA between OVX and sham control groups. Therefore, a change in the amount of ER might not participate in regulating the changes in the NO signaling and NMDA receptor function following ethanol administration in OVX rats. Instead, ER-mediated signaling pathways might be involved in the regulation, which requires to be further clarified in future research.

## 4. Materials and Methods

### 4.1. Animals

All animal care and experimental protocols were approved and carried out following the guidelines of the Institutional Animal Care and Use Committee of Tzu Chi University. Female Sprague-Dawley rats weighing 220–250 g were used in this experiment (BioLASCO Co., Ltd., Taipei, Taiwan). The rats were housed and maintained in the controlled room at 23 ± 1 °C, with 50 ± 10% humidity and a 12-h light/dark cycle.

### 4.2. Chemicals

We purchased ethanol and E2 from Sigma Co., (St. Louis, MO, USA). Zoletil^®^ 50 was purchased from Virbac Taiwan Co., Ltd. (Taipei, Taiwan). We purchased reagents used for immunohistochemistry and Western blot analysis from Sigma Co., (St. Louis, MO, USA). The reagents for electrophoresis were obtained from Bio-Rad Laboratories (Hercules, CA, USA).

### 4.3. Ovariectomy (OVX)

Under anesthetized by intraperitoneal injection of Zoletil^®^ (50 mg/kg), bilateral OVX was performed. A 1 to 1.5-cm-long midline incision through the skin and wall abdomen was made. The ovaries were exposed by following the uterine horn cranially, and a ligature (UNIK, surgical suture, USP 4/0) was made 0.5 cm before the cervix. After OVX or sham surgery, the rats were housed individually in separate cages for 4 days. On the 5th day, the rats were put back in the colony with 2 rats per cage. E2 (50 µg/kg) was administered daily by subcutaneous (s.c.) injection right after OVX until the day of the experiment.

### 4.4. Ethanol or Saline Administration

The procedure for oral administration of ethanol and saline was similar to those previously described [12]. Ethanol (3.2 g/kg, 40% *v*/*v*, 10 mL/kg) or saline (10 mL/kg) was given orally using a 16-gauge 3-inches-long stainless steel needle (Cadence Science^®^, Cadence, Inc., Staunton, VA, USA). The needle was inserted into the mouth, gently advanced along the mouth’s hard palate, and passed into the esophagus until the base of the needle. Ethanol or saline was delivered slowly.

### 4.5. Blood Pressure and Heart Rate Measurement

Blood pressure and HR measurements were performed with the tail-cuff method (CODA^TM^, Kent Scientific, Torrington, CT, USA). Seven days after OVX or sham surgery, the rats were trained for 1 h for the tail-cuff procedure for 5 consecutive days before the experiment. Rats were placed on a 40 °C heater for 30 min to achieve a tail temperature of 32–35 °C. The tail was threaded through the Occlusion Cuff to the base; the tail was then threaded again through the VPR Cuff located within 2–5 mm of the Occlusion Cuff. While securing the “Cuff Tubing”, the rats were ready for BP & HR measurement. The data was processed and counted before treatment and at different times after ethanol administration.

### 4.6. Western Blot Analysis

The procedure for Western blot analysis of brain tissue was similar to that described in earlier studies [12]. Rats were decapitated 60 min after oral gavage of ethanol or saline. Brains were rapidly removed and soaked in ice-cold Krebs solution for 1 min. The brainstems were isolated from the brain and quickly frozen by cold spray (FREEZE 75; CRC Industry Europe NV, Zele, Belgium). We cut a coronal section (1–2 mm thick) 0.5–1.5 mm rostral to the obex and isolated the area of RVLM and NA using a puncher. The isolated tissues were frozen in liquid nitrogen and stored at −85 °C. The tissue was homogenized in PRO-PREPTM solution (iNtRON Biotechnology, Republic of Korea) with a homogenizer at 10,000 rpm for 10 s. SDS was added to the sample to a final concentration of 0.1%, and 10–15 µg of protein was electrophoresed on 8.5% denaturing polyacrylamide gels. Separated proteins were transferred to the PVDF transfer membrane and probed with primary antibody, rabbit anti-nNOS polyclonal antibody (1:1000, Cell Signaling Technology, Danvers, MA, USA), rabbit anti-iNOS polyclonal antibody (1:1000, ab3523, Abcam, Cambridge, UK), and rabbit anti-eNOS monoclonal antibody (1:1000, D9A5L, Cell Signaling Technology, Danvers, MA, USA), rabbit anti-ERα polyclonal antibody (1:1000, C1355, 3231918, Sigma-Aldrich, Saint Louis, MO, USA), and rabbit anti-ERβ polyclonal antibody (1:1000, ab3576, Abcam, Cambridge, UK). Bound antibody was incubated with goat anti-rabbit IgG (1:3000, Cell Signaling Technology, Danvers, MA, USA) conjugated to horseradish peroxidase, which was reacted with Western Lighting^®^ Plus-ECL Reagent (PerkinElmer, Waltham, MA, USA). The chemiluminescent signal was digitalized by UVP Biospectrum 810 (UVP, LLC, Upland, CA, USA), and the bands were analyzed with VisionWorks LS software version 8.20 for Windows (UVP, Upland). Protein concentrations were determined by BCA method (Sigma-Aldrich, Saint Louis, MO, USA).

### 4.7. Immunohistochemistry

The procedure for immunostaining of tissue sections was similar to those previously described [42,43]. The rats anesthetized with urethane (1.2 g/kg, i.p.) were sacrificed 40 min following saline or ethanol administration. The rats were perfused intracardially with 0.9% saline followed by 300–400 mL 4% paraformaldehyde in 0.1 M PBS buffer. Coronal brainstem sections (30 μm) were prepared using a cryostat (Leica CM3050S). The sections were treated with 3% hydrogen peroxide to eliminate endogenous peroxidase activity and incubated in 2 % normal goat serum and 0.3% Triton X-100 to block non-specific binding. The sections were incubated in rabbit anti-GluN1 polyclonal antibody (1:500, Invitrogen, Waltham, MA, USA), rabbit anti-phospho-GluN1 antisera (serine 896, 1:500, Bioss, Woburn, MA, USA), rabbit anti-GluN2B polyclonal antibody (1:200, Invitrogen, Waltham, MA, USA), or rabbit anti-phospho-GluN2B antisera (serine 1480, 1:100, Bioss, Woburn, MA, USA) for 2 days at 4 °C. All sections were then incubated in biotinylated goat-anti-rabbit IgG (1:200, Vector Laboratories, Newark, CA, USA), followed by incubation for an additional 60 min with an avidin-biotin complex solution (1:50, ABC Elite, Vector Laboratories, Newark, CA, USA). Following washing, sections were reacted with a DAB substrate kit (Vector Laboratories, Newark, CA, USA) to enable visualization of the precipitate. The sections were washed and mounted on gelatin-subbed slides, and the slides were dried, dehydrated in alcohol (70–100% gradually), cleared in xylene, and cover-slipped. The sections were examined under a brightfield microscope at 100× to localize neurons in the RVLM area. Each image field was captured using a camera mounted on a microscope (Nikon Eclipse E800). A standardized region of interest (square side 600 μm) was aligned and consecutively centered on the RVLM area. The number of IR neurons on both sides of each section was counted and averaged. We randomly took 4–6 sections from each animal and calculated the average number of IR neurons per section representing the animal’s immunoreactivity to the antibody.

### 4.8. Determination of Total Nitrate-Nitrite

Nitric oxide values are represented by the total amount of nitrate and nitrite expressed in nmol/mg protein. The rats were sacrificed 60 min following saline or ethanol administration. We isolated the RVLM and NA areas from the brainstem as experiments for western blot. Total nitrate-nitrite levels were measured by nitrate-nitrite colorimetric assay kit (ThermoFisher Scientific, Bender MedSystems GmbH, Vienna, Austria) based on the Griess method, and the absorbance at 545 nm was detected.

### 4.9. Statistical Analysis

Data were presented as mean ± SEM and were plotted and analyzed statistically with GraphPad Prism version 9.00 for Windows, GraphPad Software (San Diego, CA, USA). The changes in BP and HR at different times after administration among groups were analyzed using two-way ANOVA followed by Tukey’s post-test. The results of Immunostaining, Western blot analysis, and total nitrate-nitrite were analyzed by unpaired *t*-test (comparison of two groups) or one-way ANOVA (comparison of three groups). *p* < 0.05 was considered statistically significant.

Ethics approval and consent to participate: All experimental procedures were approved and carried out in accordance with the protocol approved by the Institutional Animal Care and Use Committee of Tzu Chi University, Hualien, Taiwan (Protocol No. 106057, 108073).

## 5. Conclusions

The estrogen depletion by OVX potentiated the tachycardia accompanied by decreased pGluN1-serine 896 and lower NO levels in the RVLM following ethanol administration. Chronic E2 treatment rescued the worsened tachycardia and restored the NO levels in the RVLM. Neither OVX nor chronic E2 administration affected the protein expression of NOS and ERs. These results suggest that low estrogen levels might worsen ethanol-induced tachycardia by potentiating the inhibitory effects of ethanol on NMDA receptor function and NO level in the RVLM. This finding may provide insight into the tachycardia elicited by ethanol in menopausal women or women taking anti-estrogen medication.

## Figures and Tables

**Figure 1 ijms-24-05087-f001:**
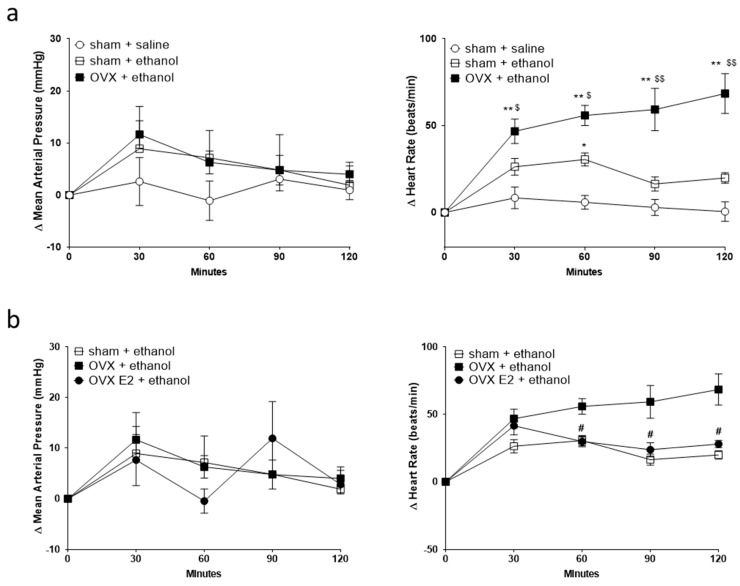
Line graphs show the time course of changes in mean arterial pressure (MAP, left panel) and heart rate (HR, right panel) after oral gavage of normal saline (*n* = 6) or ethanol (3.2 g/kg) in sham (*n* = 5) and ovariectomy (OVX, *n* = 5) rats (**a**), and OVX rats daily supplemented with estradiol (OVX E2, *n* = 6) (**b**). The value of MAP and HR 30 min before saline or ethanol administration is taken as a baseline (0). Values are means ± SEM. * *p* < 0.01, ** *p* < 0.001 compared with the corresponding time point in the sham + saline group; $ *p* < 0.05, $$ *p* < 0.001 compared with the corresponding time point in the sham + ethanol group; # *p* < 0.01 compared with the corresponding time point in the OVX + ethanol group as determined by two-way ANOVA followed by Tukey post-test.

**Figure 2 ijms-24-05087-f002:**
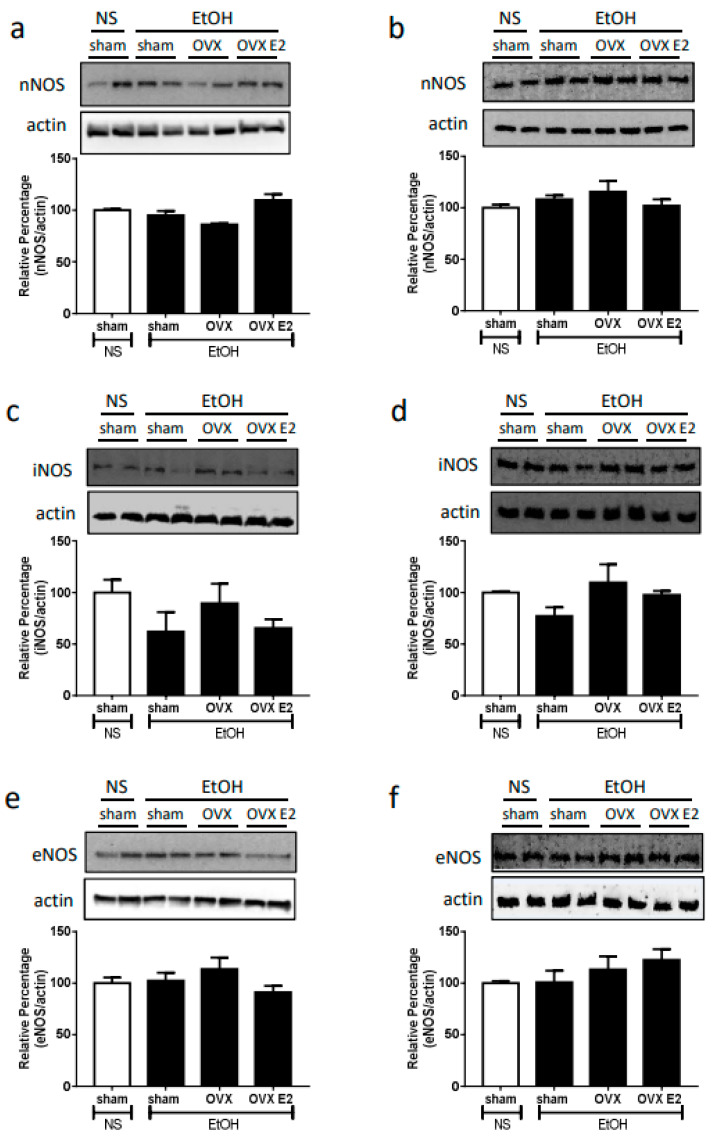
The top panels show representative western blot analysis; the bottom graphs show the percentage changes in the expression of nNOS, iNOS, and eNOS in RVLM (**a**,**c**,**e**) and nucleus ambiguus (NA, (**b**,**d**,**f**)) at 60 min after oral gavage of normal saline (NS, 10 mL/kg) or ethanol (EtOH, 3.2 g/kg) in different treatment groups (*n* = 4 each). The ratio of different NOS to β-actin in the saline group is taken as a control of 100%.

**Figure 3 ijms-24-05087-f003:**
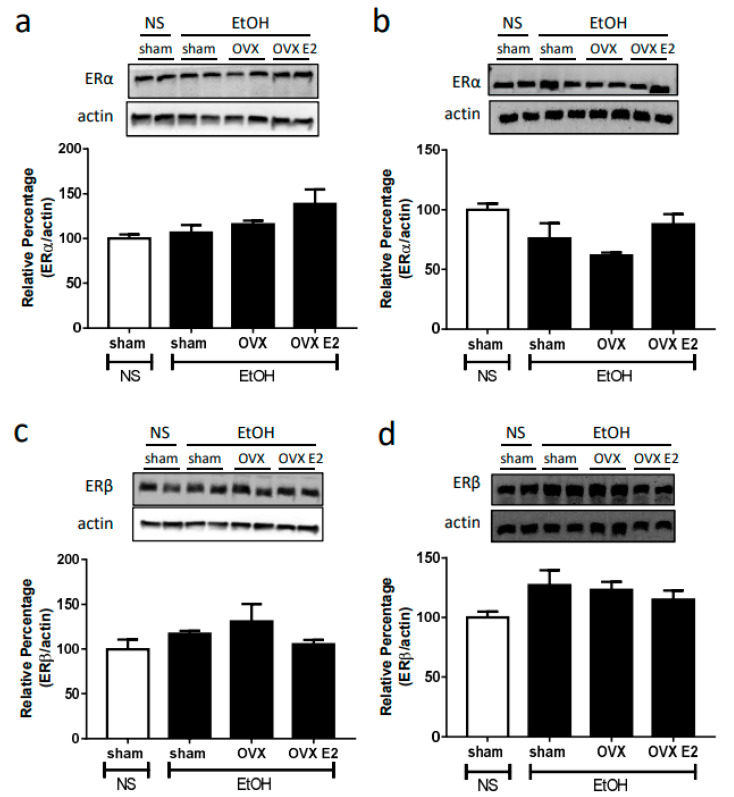
The top panels show representative western blot analysis; the bottom graphs show the percentage changes in the expression of ERα and ERβ in RVLM (**a**,**c**) and NA (**b**,**d**) at 60 min after oral gavage of normal saline (NS, 10 mL/kg) or ethanol (EtOH, 3.2 g/kg) in different treatment groups (*n* = 4 each). The ratio of different NOS to β-actin in the saline group is taken as a control of 100%.

**Figure 4 ijms-24-05087-f004:**
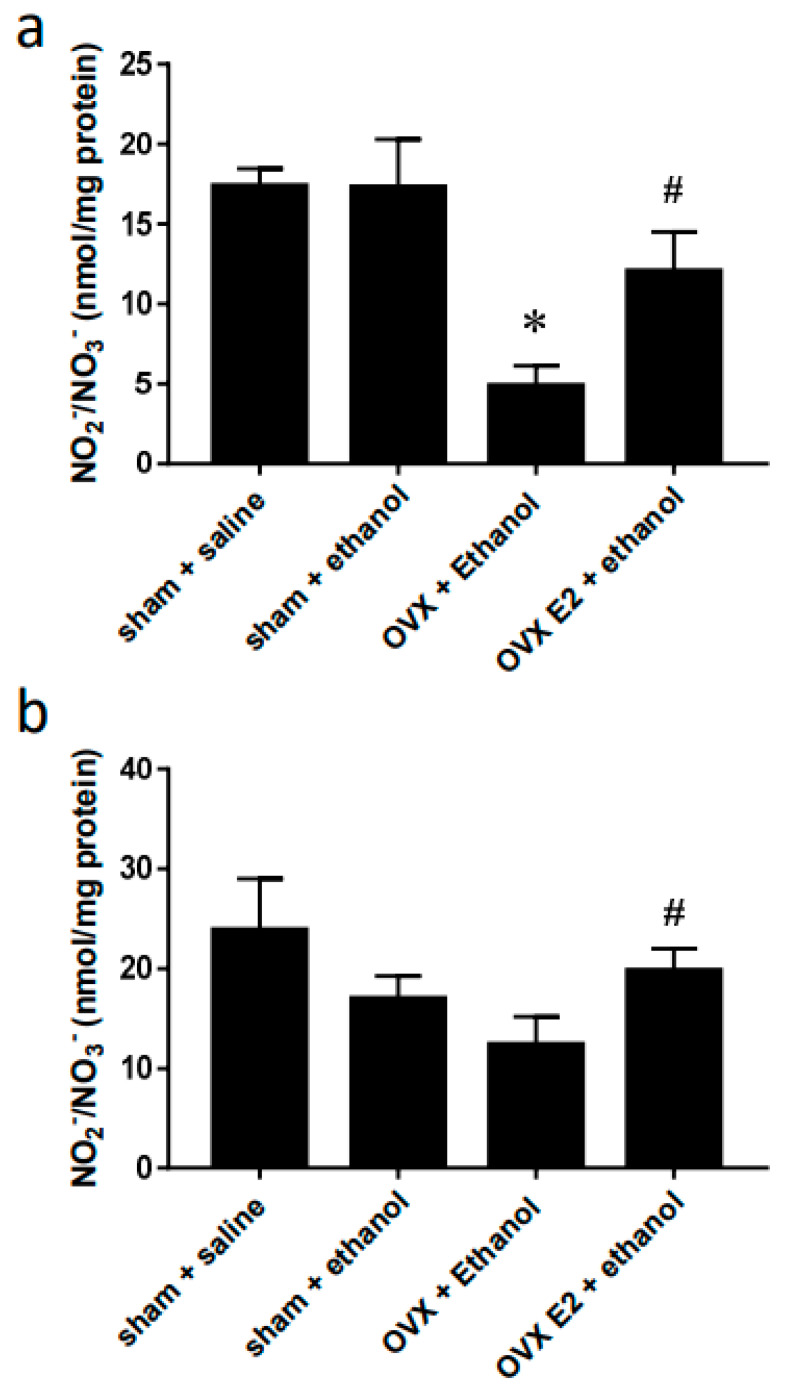
Bar graphs show changes in NOx levels in RVLM (**a**) and NA (**b**) at 60 min after oral gavage of normal saline (10 mL/kg) or ethanol (3.2 g/kg). Values are mean ± SEM. Data were analyzed by one-way ANOVA followed by Tukey’s post-hoc test. * *p* < 0.05 compared with sham + ethanol, # *p* < 0.05 compared with OVX + ethanol. (*n* = 4).

**Figure 5 ijms-24-05087-f005:**
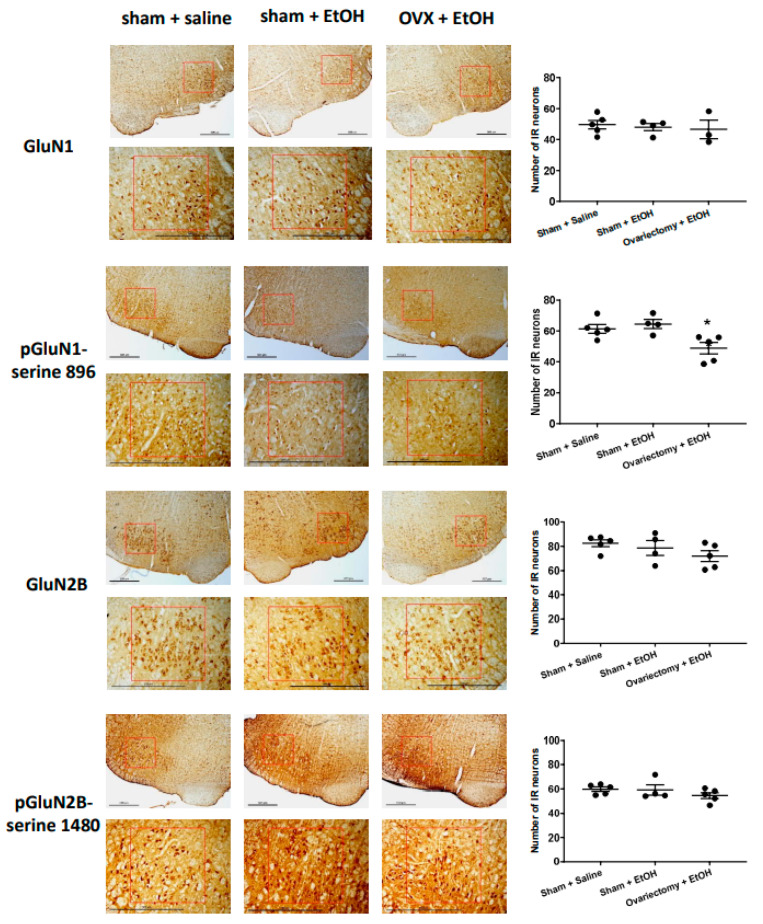
Left panel: representative immunohistochemical analysis of the expression of NMDA receptor GluN1 subunit (GluN1), phosphoserine 896 on NMDA receptor GluN1 subunit (pGluN1-serine 896), NMDA receptor GluN2B subunit (GluN2B) and phosphoserine 1480 on NMDA receptor GluN2B subunit (pGluN2B-serine 1480) in the RVLM area at 40 min following oral administration of saline (10 mL/kg) or ethanol (EtOH) (3.2 g/kg, 40% *v*/*v*, 10 mL/kg) in sham and OVX rats. Scale bar equals to 600 μm. The red box indicates the region of the RVLM. Right panel: the bar graph shows the number of immunoreactive (IR) neurons in the expression of the GluN1, pGluN1-serine 896, GluN2B, pGluN2B-serine 1480 following the application of saline or ethanol (*n* = 3–5, each). The average number of IR neurons in the RVLM area in rats showed no changes in the immunoreactivity of GluN1, GluN2B, and pGluN2B-serine 1480, but a significant decrease in pGluN1-serine 896 among different groups. Values are mean ± SEM. * *p* ≤ 0.05 compared to the sham + saline and sham + EtOH groups analyzed by one-way ANOVA followed by Bonferroni’s test.

## Data Availability

Data available upon reasonable request.

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
