# Peer review of "Ovariectomy Exacerbates Acute Ethanol-Induced Tachycardia: Role of Nitric Oxide and NMDA Receptors in the Rostral Ventrolateral Medulla"

_ijms, 2023, doi:10.3390/ijms24065087_

Round 1
Reviewer 1 Report
1. To clarify the role of ovary hormone in regulation of NMDA receptor in Fig. 5, the authors should examine whether the treatment with estradiol could reverse ethanol-induced decrease in the number of neurons expressing phospho-GluN1 (serine 896) in RVLM of ovariectomized rats.
2. Which isoform of NOS may be involved in the effect of ethanol?
3. What is the causal relation between NMDA receptor activation and NOS activation in ethanol-induced tachycardia in ovariectomized rats? Pharmacological inhibiton of NMDA receptor and NOS is suggested to be used for answering this question.
4. A typo is appeared in the title of Y-axis.
Author Response
- To clarify the role of ovary hormone in regulation of NMDA receptor in Fig. 5, the authors should examine whether the treatment with estradiol could reverse ethanol-induced decrease in the number of neurons expressing phosphoGluN1 (serine 896) in RVLM of ovariectomized rats
Thank you for your comment. We demonstrated in the immunohistochemical study that ethanol administration decreased pGluN1-serine 896 positive neurons in the RVLM of OVX rats compared with sham rats, even though we did not conduct estradiol supplements of OVX rats. This result provided evidence that OVX might modulate ethanol effects on NMDA receptor function by regulating NMDA receptor phosphorylation (pGluN1-serine 896). The results would give a possible relationship between NMDA receptor function and NO levels, which might support part of the conclusion. However, it is worth mentioning that the results cannot reveal the causal relationship between NMDA receptor function and NO signals in OVX regulation of ethanol effects. The primary hormones produced by the ovaries include estrogen and progesterone. The specific role of estrogen and/or progesterone in regulating NMDA receptor phosphorylation needs to be further clarified. We have noticed this limitation and modified the discussion accordingly (P. 13).
- Which isoform of NOS may be involved in the effect of ethanol?
Thank you for your comment. In our previous study (Situmorang et al., 2018), we demonstrated that 8 days of ethanol administration resulted in a decrease in nNOS level. However our current study found that a single ethanol administration did not affect any of the three types of NOS suggesting that acute ethanol consumption may not alter NOS expression. Nonetheless, our findings indicate that the NO level was lower in the OVX group, implying a decrease in NOS activity instead of its expression. Given that nNOS is the primary type of NOS in the brain, we speculate that it may be more closely associated with the effect of OVX and ethanol use. We have included this in the discussion (P. 12).
- What is the causal relation between NMDA receptor activation and NOS activation in ethanol-induced tachycardia in ovariectomized rats? Pharmacological inhibition of NMDA receptor and NOS is suggested to be used for answering this question.
Thank you for your thoughtful comment. The causal relationship between NMDA receptor and nNOS activation in several neuronal functions has been carried out in previous studies. Our findings indicated that PKC-regulated NMDA receptor function (pGluN1-serine 896) and NO levels in the RVLM might contribute to OVX regulation of ethanol-induced tachycardia. Thus, by pharmacological approach, administration of NMDA receptor and NO signals modulators into the RVLM might regulate the ethanol-induced tachycardia, which might further prove the causal relation between NMDA receptor and NO signaling in OVX regulation of ethanol effects. However, microinjecting the modulators into the RVLM is technically tricky in awake rats used in the current study. We have acknowledged this in our discussion (P. 13).
- A typo is appeared in the title of Y-axis.
We have corrected the typo errors in the title of Y-axis in Figure 1a. (P. 6)
Reviewer 2 Report
Thank you for giving me the opportunity to review this article. In this study, the authors conducted an animal experiment using female rats to examine the relationship between cardiovascular abnormality caused by alcohol consumption and female sex hormone. Ethanol consumption increased the heart rate, which was further increased in ovariectomized rats. And these phenomena were improved by female sex hormone replacement. The authors focused on the amount of nitric acid and nitrite in RVTM and discussed the relationship between female hormones and ethanol-induced tachycardia. These results may be meaningful for a better understanding the etiology and treatment of alcohol-induced tachycardia in women. The followings are my comments and questions. -In this study, the ethanol concentration and volume were fixed only to 3.2 g/kg, 40% v/v, 10 mL/kg. Therefore, readers cannot see dose-dependent changes in the results. Please describe the reason why the authors selected this alcohol concentration and volume. -According to the Instruction for authors, "Abbreviations should be defined the first time they appear in each of three sections: the abstract; the main text; the first figure or table." The words, E2, GABA, and MAP, were not defined at the first time. The words, RVLM and OVX, were defined several times. -At the head of the fourth paragraph in the Introduction, the authors described, "In conscious female rats, OXV abolished the ------." Is the word "OXV" a miss-written for "OVX"? -In the Results, the subcategory of "3.1" was used twice.Author Response
- In this study, the authors conducted an animal experiment using female rats to examine the relationship between cardiovascular abnormality caused by alcohol consumption and female sex hormone. Ethanol consumption increased the heart rate, which was further increased in ovariectomized rats. And these phenomena were improved by female sex hormone replacement. The authors focused on the amount of nitric acid and nitrite in RVTM and discussed the relationship between female hormones and ethanol-induced tachycardia. These results may be meaningful for a better understanding the etiology and treatment of alcohol-induced tachycardia in women.
Thank you very much for summarizing our findings.
- In this study, the ethanol concentration and volume were fixed only to 3.2 g/kg, 40% v/v, 10 mL/kg. Therefore, readers cannot see dose-dependent changes in the results. Please describe the reason why the authors selected this alcohol concentration and volume.
In the present study, we administered rats with ethanol by oral gavage at a dose of 3.2 g/kg (40% v/v). Our previous study (Situmorang et al., 2018) has shown that the blood concentration was around 80-100 mg/dL (0.08-0.1%) 30-60 min following oral gavage of ethanol (3.2 g/kg), which is comparable to levels observed in many human drinkers. The above statements have been added in the Discussion section (P. 12).
- According to the Instruction for authors, "Abbreviations should be defined the first time they appear in each of three sections: the abstract; the main text; the first figure or table." The words, E2, GABA, and MAP, were not defined at the first time. The words, RVLM and OVX, were defined several times.
We have re-checked the abbreviation in the TEXT.
- At the head of the fourth paragraph in the Introduction, the authors described, "In conscious female rats, OXV abolished the ------." Is the word "OXV" a misswritten for OVX"?
We have corrected the typo errors (P. 2).
- In the Results, the subcategory of "3.1" was used twice.
We have corrected the format (P. 6).
Reviewer 3 Report
The manuscript presents findings of research on the Ovariectomy Exacerbates Acute Ethanol-Induced Tachycardia: Role of Nitric Oxide and NMDA Receptors in the rostral ventrolateral medulla. The paper presents very interesting results as well as an inquisitive and reliable interpretation of the research results. The methodology adequately described and conclusion consistent with the evidence and arguments presented.
Author Response
The manuscript presents findings of research on the Ovariectomy Exacerbates Acute Ethanol-Induced Tachycardia: Role of Nitric Oxide and NMDA Receptors in the rostral ventrolateral medulla. The paper presents very interesting results as well as an inquisitive and reliable interpretation of the research results. The methodology adequately described and conclusion consistent with the evidence and arguments presented.
Thanks you very much for the review comments.
Round 2
Reviewer 1 Report
no further comment on the manuscript